# Control and Modification of Nanostructured Materials by Electron Beam Irradiation

**Shun-Ichiro Tanaka**

Micro System Integration Center (μSIC), Tohoku University, Aramaki, Sendai 980-0845, Japan; shunichiro.tanaka.d2@tohoku.ac.jp

**Abstract:** I have proposed a bottom-up technology utilising irradiation with active beams, such as electrons and ions, to achieve nanostructures with a size of 3–40 nm. This can be used as a nanotechnology that provides the desired structures, materials, and phases at desired positions. Electron beam irradiation of metastable θ-$Al_2O_3$, more than $10^{19}$ e/$cm^2$s in a transmission electron microscope (TEM), enables the production of oxide-free Al nanoparticles, which can be manipulated to undergo migration, bonding, rotation, revolution, and embedding. The manipulations are facilitated by momentum transfer from electrons to nanoparticles, which takes advantage of the spiral trajectory of the electron beam in the magnetic field of the TEM pole piece. Furthermore, onion-like fullerenes and intercalated structures on amorphous carbon films are induced through catalytic reactions. δ-, θ-$Al_2O_3$ ball/wire hybrid nanostructures were obtained in a short time using an electron irradiation flashing mode that switches between $10^{19}$ and $10^{22}$ e/$cm^2$s. Various α-$Al_2O_3$ nanostructures, such as encapsulated nanoballs or nanorods, are also produced. In addition, the preparation or control of Pt, W, and Cu nanoparticles can be achieved by electron beam irradiation with a higher intensity.

**Keywords:** electron irradiation; excited reaction field; transmission electron microscope; nanomaterials; manipulation; nanostructure; Al; $Al_2O_3$





## 1. Introduction

The size range of several tens of nanometres represents a transition region from "top-down" to "bottom-up" processes in nanotechnology. The author has proposed a bottom-up technology utilising active beam irradiation to achieve nanostructures with a size of 3–40 nm. Recent developments in focusing and scanning technologies for active beams, such as electrons or ions, in tabletop apparatuses enable the evolution and control of various types of nanostructures, which can provide desirable hybrid structures, materials, and phases at the desired positions on the nanometre scale.

The source of the electron beam used by the author's group is a transmission electron microscope (TEM) equipped with either an $LaB_6$ filament or a field emission gun. Although TEM has been widely used as an analysis tool for studying nanostructure and element distributions, we consider a specimen stage of 3 mm in diameter as a reaction field, and focused electrons with an intensity more than 50 times higher than that used under normal observation conditions. The electron irradiation intensity ranged from $5 \times 10^{19}$ to $4 \times 10^{23}$ e/$cm^2$s ($8 \times 10^4$–$6 \times 10^8$ A/$m^2$) in our experiment, and we succeeded in producing oxide-free nanoparticles via electron irradiation of the oxide and facilitated their subsequent manipulation.

In this paper, three topics on the manipulation and control of ceramics and metals by electron beam irradiation are discussed: (1) the preparation of Al nanoparticles and their nanostructure evolution starting from metastable $Al_2O_3$, (2) their manipulation by electrons, and (3) the effects of electron irradiation on other nanoparticles, such as Cu, Pt, and W, to prepare nanofilms. This review is based on papers published between 1995 and 2005.

## 2. Electron Excited Reaction Field

### 2.1. Overview

In the "Exploratory Research for Advanced Technology" (ERATO) "Tanaka Solid Junction Project," JST [1], held in 1993–1998, I commenced an innovative challenge to fabricate nano-/microstructures by irradiation with an energy beam such as electrons and ions, based on the proposed concept of an "excited reaction field." One of the characteristics of such field is that the beams are obtained on the specimen stage of a TEM for electrons, and on the milling/thinning stage for ions. In other words, observation or specimen preparation apparatuses are utilised for their beam source and excited reaction fields. Irradiation with these energy beams has the following merits: first, it facilitates the selection of the site/energy/reaction. Second, it can induce nonequilibrium/catalytic reactions, making it possible to manipulate atomic clusters, synthesise nanomaterials, control the nanostructure and phase, and modify the nanospace.

### 2.2. Effects of Electron Beam Irradiation on $\theta$-$Al_2O_3$

The normal electron beam intensity for observation by a TEM equipped with an $LaB_6$ filament is on the order of $10^{18}$ e/cm$^2$s (1600 A/m$^2$) for bright-field imaging and $5 \times 10^{16}$ e/cm$^2$s s (80 A/m$^2$) for selected area diffraction measured by a fluorescent plate. We increased the electron beam density by increasing the current in the condenser lens to between $10^{19}$ and $10^{22}$ e/cm$^2$s to enable electron irradiation of the nanomaterials.

When an electron beam irradiates metastable $\theta$-$Al_2O_3$ particles at an intensity of $10^{20}$ e/cm$^2$s in a TEM, successive reactions occur: ① Al nanoparticle inducement, ② rotation, revolution, and migration to coalesce/embed, and ③ formation of onion-like fullerenes and Al intercalation, as shown in Figure 1. Furthermore, ④ $\theta$- or $\delta$-$Al_2O_3$ nanostructures are obtained under flashing-mode irradiation (rapid switching between $10^{19}$ and $10^{22}$ e/cm$^2$s).

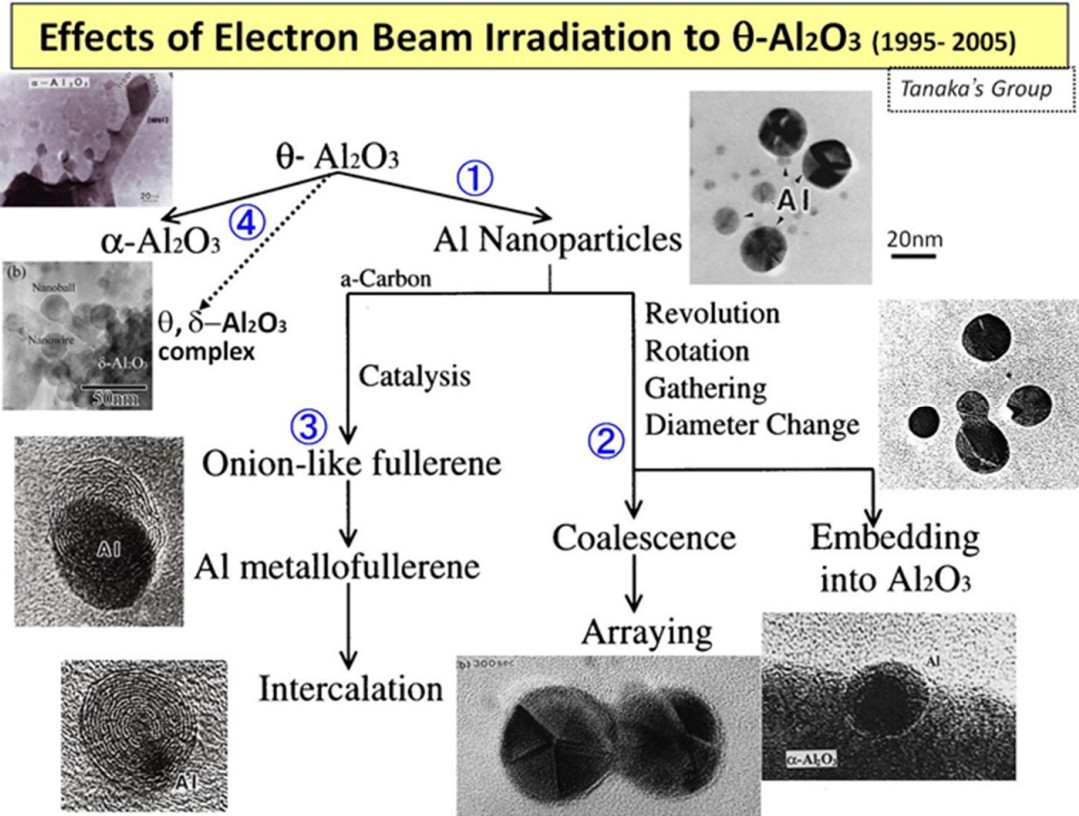

**Figure 1.** Evolution of the reactions and nanostructures induced by electron beam irradiation of metastable $\theta$-$Al_2O_3$ in TEM. Numbers ①–④ denote the reaction routes explained [1].

## 3. Nanostructure Evolution

### 3.1. Al Nanoparticles

The starting material consists mainly of a metastable θ-$Al_2O_3$ monoclinic structure accompanied by orthorhombic δ-$Al_2O_3$ ($Al_2O_3$ allotrope), and it is different from stable α-$Al_2O_3$ with a trigonal structure. The θ-$Al_2O_3$ powder was synthesised using the vaporised metal combustion method (Admatechs Company Limited) [2].

Electron irradiation ($2.1 \times 10^{20}$ e/$cm^2$s for 50 s) of one θ-$Al_2O_3$ particle with a diameter of 100 nm on the φ3 mm specimen stage of the TEM led to the formation of Al nanoparticles with a diameter of 2–20 nm and a stable α-$Al_2O_3$ particle, as shown in Figure 2 [3]. The TEM used was a JEOL JEM-2010 equipped with an $LaB_6$ filament and a specimen chamber under a vacuum of $10^{-7}$ Pa, obtained by a direct coupling sputter ion pump. Under these conditions, the electrons cut the Al-O bond in θ-$Al_2O_3$, which was followed by the loss of oxygen to the vacuum and Al atom recombination to form nanoparticles, as schematically shown in Figure 3. In the electron-irradiated area, transformation or rearrangement from θ-$Al_2O_3$ to stable α-$Al_2O_3$ also occurred.

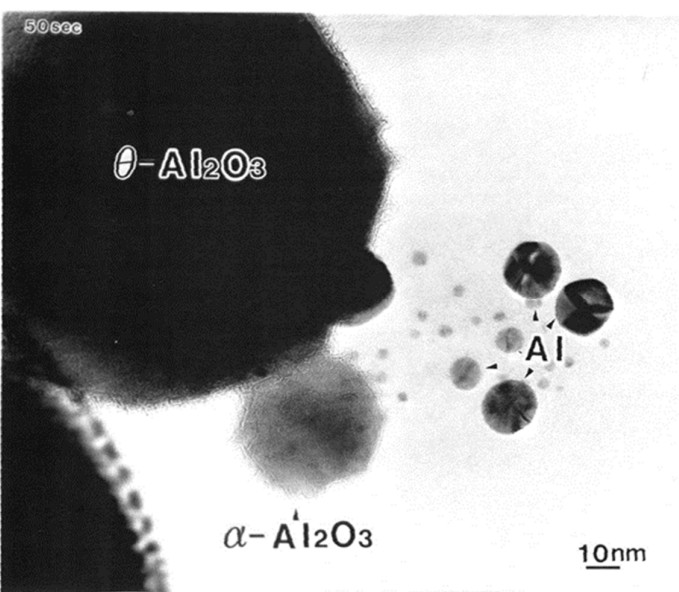

**Figure 2.** Al nanodecahedra of 10–17 nm in diameter and α-$Al_2O_3$ nanoparticles obtained by electron irradiation of metastable θ-$Al_2O_3$ with an intensity of $2.1 \times 10^{20}$ e/$cm^2$s in TEM [3].

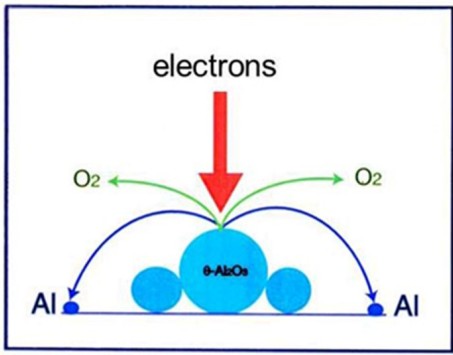

**Figure 3.** Schematic of Al nanodecahedra formation by electron irradiation in a TEM. Electrons cut the Al-O bond of metastable θ-$Al_2O_3$, which decomposes into Al and O atoms, and finally the Al atoms recombine to form twinned decahedra surrounded by {111} surfaces.

θ- and δ-$Al_2O_3$ appear as low-temperature phases in the allotropic transformation from γ-$Al_2O_3$ to α-$Al_2O_3$ upon heating. The starting powder used in this experiment was obtained by melting Al metal powder, vaporisation, collision of droplets, and quenching

into the metastable phase [2]. The powder particles had a spherical shape with an average diameter of approximately 10 μm. No stable α-$Al_2O_3$ structure was observed by X-ray diffraction, and an equilibrium thermodynamic consideration was invalid for metastable θ-$Al_2O_3$, where the binding energy for Al-O was lower than that for α-$Al_2O_3$. The reaction was expected to proceed via a nonequilibrium route. Electron irradiation promoted the decomposition of metastable θ-$Al_2O_3$, recombination of Al atoms, loss of a part of oxygen atoms into vacuum, and transformation to stable α-$Al_2O_3$, as shown in Figure 3, which resembles one stile of the electron-stimulated desorption.

Al nanoparticles have a twinned decahedron structure surrounded by {111} surfaces, which has been reported in typical face-centred cubic noble metals, such as Au, Pt, or Ag, whereas no report has been published on Al because of its easily oxidised surface. Decahedra appeared with diameters in the range of 10–20 nm, and further electron irradiation of a set of nanodecahedra enabled their manipulation, as discussed in the next section.

*3.2. Manipulation*

In the ERATO project, TEM has been used not only as an observation apparatus but also as a tool for the manipulation of nanostructures by electrons on the specimen stage. The 200 keV JEM-2010 TEM has a pole piece with a magnetic field of $10^4$ Gauss from top to bottom, where the specimen stage is located slightly above the centre. Based on the kinetic features analysed by Horiuchi et al. [4], the momentum transfer from electrons to nanoparticles is the source of their movement. The spiral trajectory of the electrons causes Al nanoparticles on the specimen stage to experience both a tangential force to rotate and revolve and a centripetal force to migrate, bond, and embed. The features around the specimen stage and the driving forces are illustrated in Figure 4.

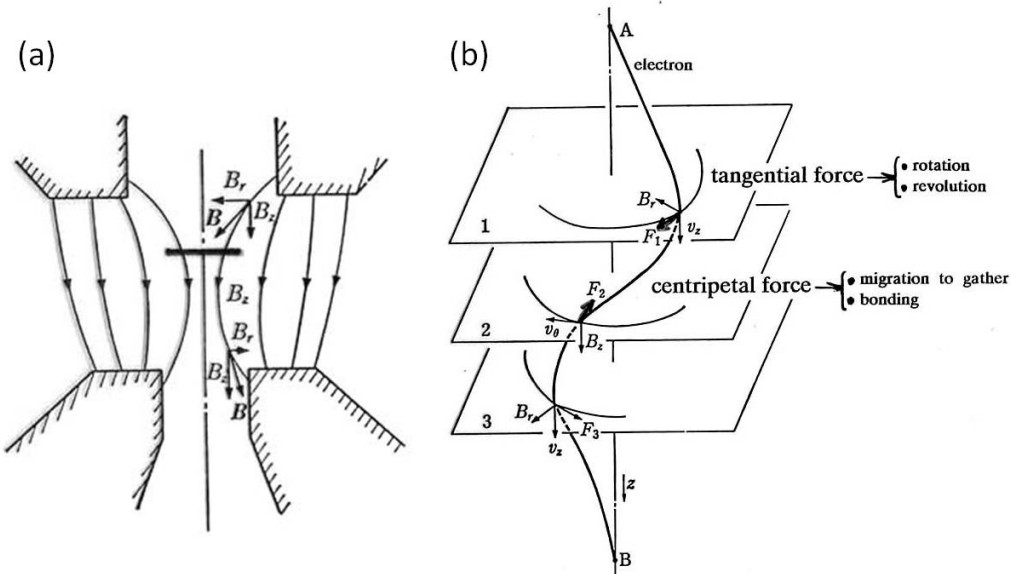

**Figure 4.** Model of the interaction between electrons and particles on the specimen stage in TEM. The magnetic field inside a pole piece is $10^4$ Gauss in a 200 keV TEM. (**a**) Electrons follow a spiral trajectory in the magnetic field to transfer forces or momentum such that (**b**) the tangential force results in rotation and revolution and the centripetal force induces migration to gather, bond, and embed the particles.

3.2.1. Migration and Bonding

A set of Al nanodecahedra migrated and bonded to the irradiation centre of the electron beam, as shown in Figure 5. Analysis by superposition before and after irradiation, as shown in Figure 5c, revealed that migration, diameter decrease, revolution, and bonding occurred in the course of irradiation over 1200 s at $2.1 \times 10^{20}$ e/$cm^2$s s, which was mainly due to the centripetal force, as shown in Figure 4. The bonding step of the two nanoparticles

is shown in Figure 6, where the (111) planes are aligned parallel by rotation before necking (a) and then necking with Σ3 twinning (b) to eventual coalescence (c) [5–7].

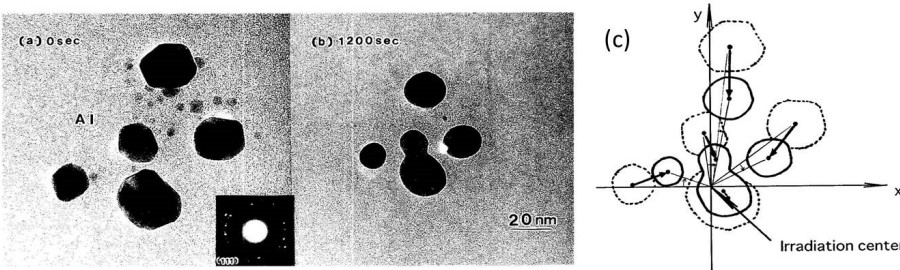

**Figure 5.** Effects of electron irradiation of a set of Al nanodecahedra inducing migration to the irradiation centre, as well as revolution and rotation. The intensity was $2.1 \times 10^{20}$ e/cm²s for (**a**) 0 s and (**b**) 1200 s. A schematic view of the effect is shown in (**c**) [6].

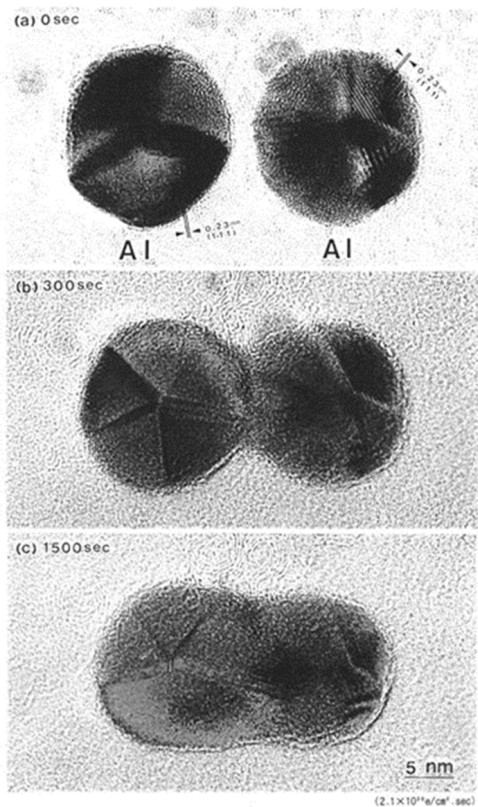

**Figure 6.** Al nanodecahedra (**a**) bonded by electron irradiation with an intensity of $2.1 \times 10^{20}$ e/cm²s after 300 s (**b**) and coalesced after 1500 s (**c**). The (111) planes in two Al nanodecahedra were aligned parallel by rotation and exhibited a Σ3 coincidence site lattice (CSL) boundary around the neck at 300 s [5,7].

### 3.2.2. Rotation and Revolution

The force acting on the nanoparticle in the magnetic field was also tangential, as well as centripetal, as shown in Figure 4, inducing a clockwise rotation of the nanoparticle on the specimen stage. The speed of rotation measured by the change in angle increased as the irradiation intensity increased for irradiation times less than 1000 s, as shown in Figure 7. The saturation of the rotation with longer exposure is considered to stem from a decrease in the diameter of the particle [6,8].

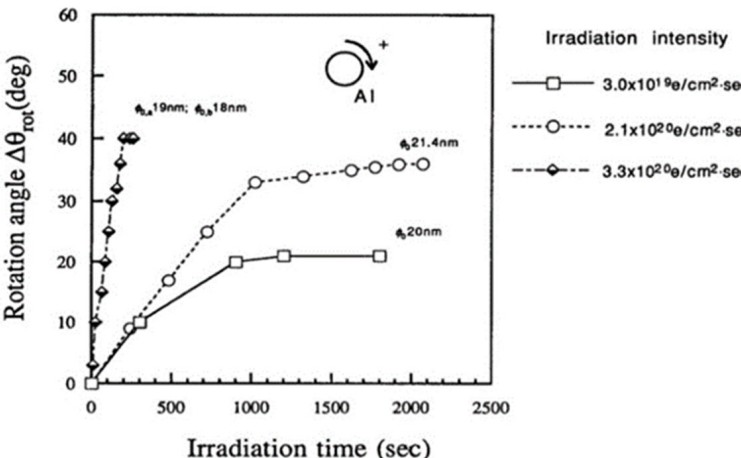

**Figure 7.** Rotation of Al nanodecahedra and its dependence on the irradiation intensity ranging from $3.0 \times 10^{19}$ to $3.3 \times 10^{20}$ e/cm$^2$s. The diameter of the nanoparticles is 20 nm, denoted on the line as $\varphi_0$ [6].

To confirm the effects of the tangential force on the nanoparticle, the magnetic direction in the pole piece was changed from bottom to top by reforming the TEM. Figure 8 shows the revolution behaviour of nanoparticles in both directions of the magnetic field. The clockwise and counterclockwise revolutions of the Al nanoparticles clearly depended on the direction of the magnetic field, and their speed increased as the irradiation intensity increased. The revolution of the nanoparticles was accompanied by their migration, as shown in Figure 5c.

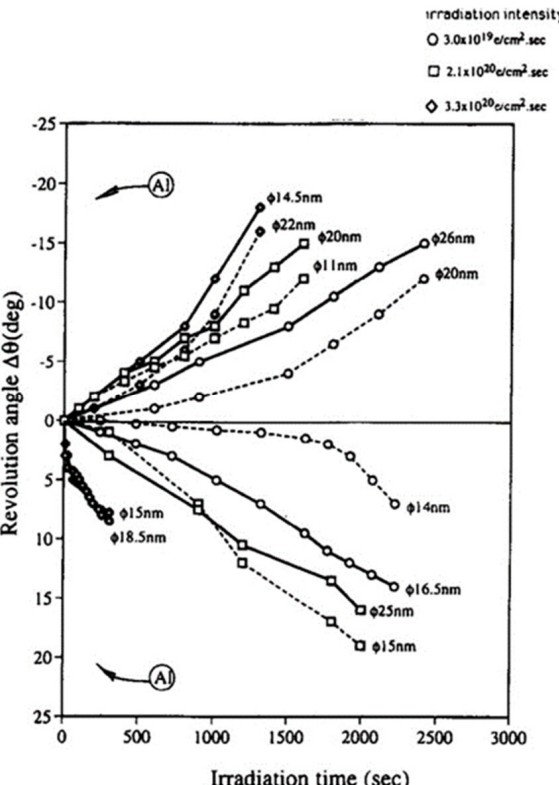

**Figure 8.** Clockwise (+) and counterclockwise (−) revolution of Al nanoparticles during electron irradiation controlled by changing the magnetic field direction. The case shown in Figure 5 induced the clockwise revolution, whereas reversal of the magnetic field (bottom to top) induced the opposite direction of revolution. Here, $\varphi$ indicates the diameter of the initial Al particle [8].

### 3.2.3. Embedding

The forces discussed in Sections 3.2.1 and 3.2.2 manipulate the nanoparticles to cause migration and embedding into the substrate. Figure 9 shows an example of an Al nanoparticle embedded in the $\alpha$-Al$_2$O$_3$ matrix following electron irradiation at $10^{20}$ e/cm$^2$s, which is attributed to an epitaxial relationship between the two substances. This technique can be utilised for the implantation of catalysts, such as Pt nanoparticles, at the desired position in a matrix [9].

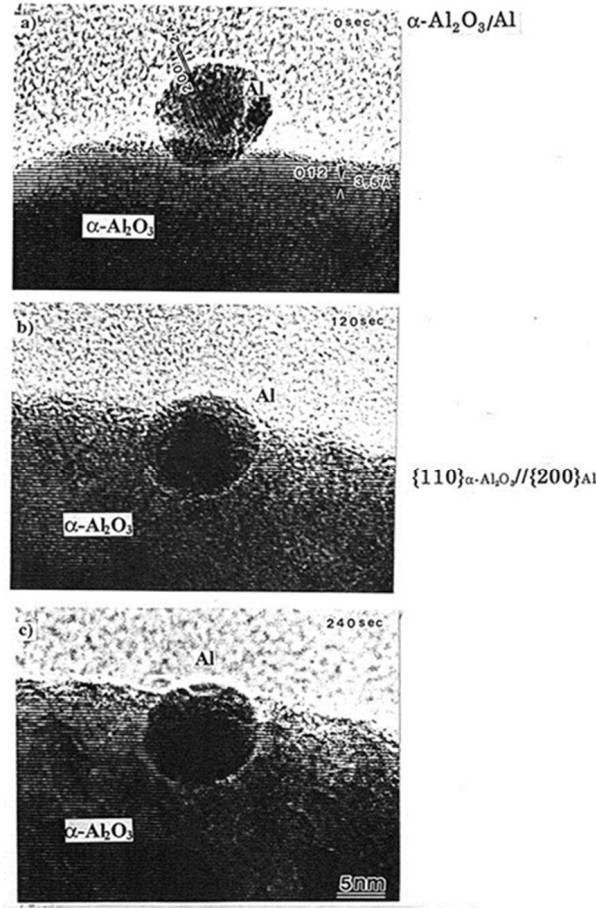

**Figure 9.** Al nanoparticle migrated and embedded into the $\alpha$-Al$_2$O$_3$ matrix by electron irradiation of the order of $10^{20}$ e/cm$^2$s for (**a**) 0 s (**b**) 120 s and (**c**) 240 s. The $\alpha$-Al$_2$O$_3$/Al interface structure exhibits a {11–20}$\alpha$-Al$_2$O$_3$//{200}Al epitaxial relationship even after electron irradiation for 240 s [9].

### 3.3. Fullerene and Intercalation

A series of nanostructures were formed on the amorphous carbon nanofilm of the specimen mesh, which was suspended on a Cu grid. Electron irradiation induced the formation of an onion-like fullerene nanostructure under the Al nanoparticles by the catalysis effect and promoted the intercalation of Al atoms between the graphite shells.

### 3.3.1. Onion-Like Fullerene

Giant onion-like fullerenes were induced from the amorphous carbon nanofilms under the Al nanodecahedra by electron beam irradiation, as shown in Figure 10. A catalytic reaction nucleated a graphitic flake along the edge of the Al nanoparticle, and prolonged electron irradiation induced the growth of the fullerene, shrinking of the nanoparticles, and intercalation of Al atoms between the shells, as shown in Figures 11 and 12 [5,10].

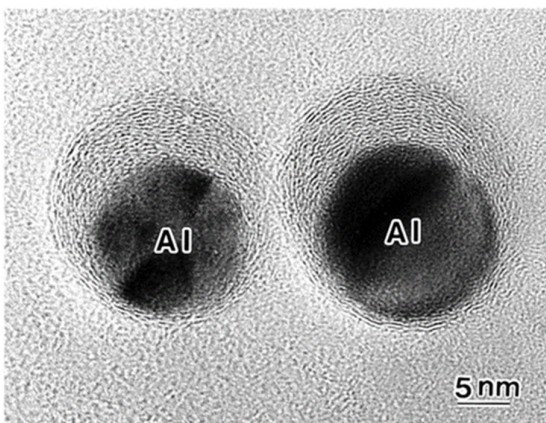

**Figure 10.** Giant onion-like fullerenes induced under Al nanoparticles by electron irradiation at an intensity of $3.0 \times 10^{19}$ e/cm$^2$s for 2050 s. The Al nanoparticles are surrounded by onion-like fullerene shells [10].

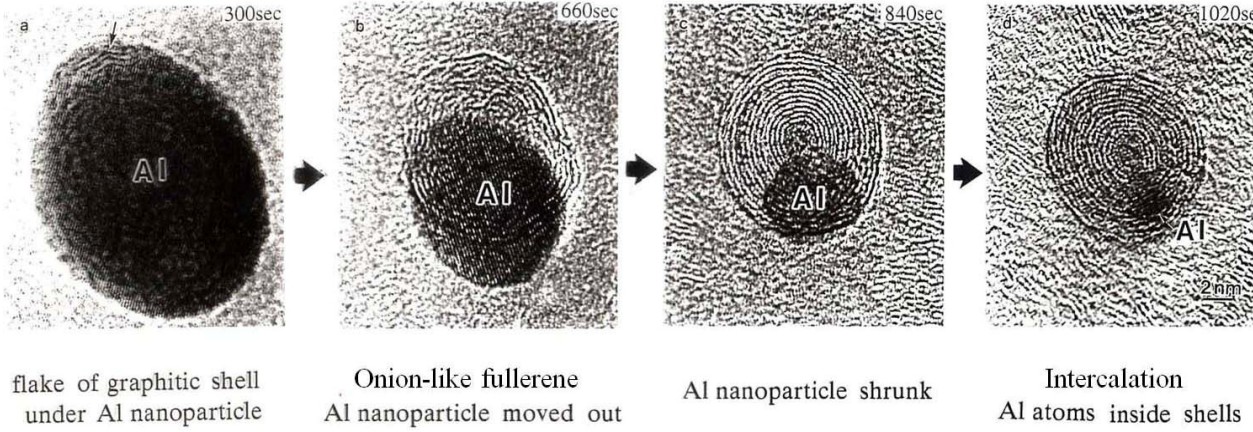

flake of graphitic shell under Al nanoparticle

Onion-like fullerene Al nanoparticle moved out

Al nanoparticle shrunk

Intercalation Al atoms inside shells

**Figure 11.** Series of reactions between Al nanoparticles and the amorphous carbon nanofilm (used as a specimen holder for TEM) induced by electron irradiation with an intensity of $10^{20}$ e/cm$^2$s. Reactions proceed from the catalytic formation of (**a**) a graphitic shell and (**b**) an onion-like fullerene under the Al nanoparticle, followed by (**c**) shrinking of the nanoparticle and (**d**) intercalation of Al atoms inside the shell [10].

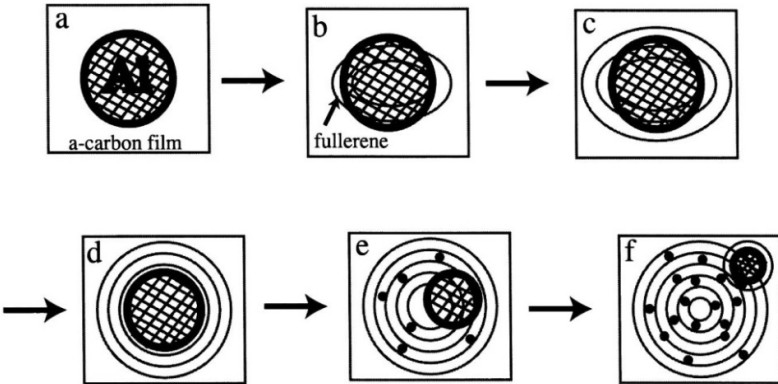

**Figure 12.** Schematic of a series of interaction behaviours between an Al nanoparticle and an amorphous carbon film under electron irradiation (**a**:initial). The following steps occurred: a nucleus of the giant onion-like fullerene was first induced under the Al particles (**b**). Al nanoparticles were encapsulated in the giant onion-like fullerene (**c**,**d**). Al nanoparticles moved outside of the giant onion-like fullerene (**e**), which also induced a new giant onion-like fullerene (**f**). Finally, intercalation progressed as Al atoms migrated inside the giant onion-like fullerene shells (**e**,**f**) [10].

### 3.3.2. Intercalation

To verify Al atom intercalation into the graphitic shells, an analysis using energy dispersive spectroscopy (EDS), for which the JEM-2010 TEM was equipped, was conducted on the electron irradiated specimen. Figure 13 shows the presence of Al atoms in the circled area of 7 nm in diameter, accompanied by C and Cu in the grid. Moreover, the expansion of the graphite (002) lattice spacing confirmed the presence of Al atoms between the layers, as shown in Figure 14. However, the growth saturated below the composition of $Al_2C_6$ owing to the blocking effect of the coexisting Cu atoms or the constraint of passing through multiple carbon layers. Electron energy loss spectra (EELS) also suggested the partial replacement of carbon atoms by Al or Cu atoms through σ bond, $sp^2$, decrease, as shown in Figure 15 [10–12].

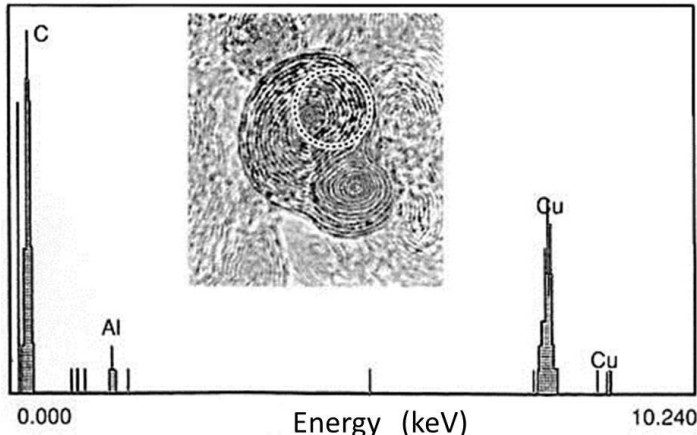

**Figure 13.** EDS analysis of the area inside the dotted circle in the nanostructure shown in Figure 11. The electron probe size was 7 nm in diameter. An Al peak was detected, indicating the possibility of intercalation, whereas the Cu peaks came from the Cu grid of the membrane [10].

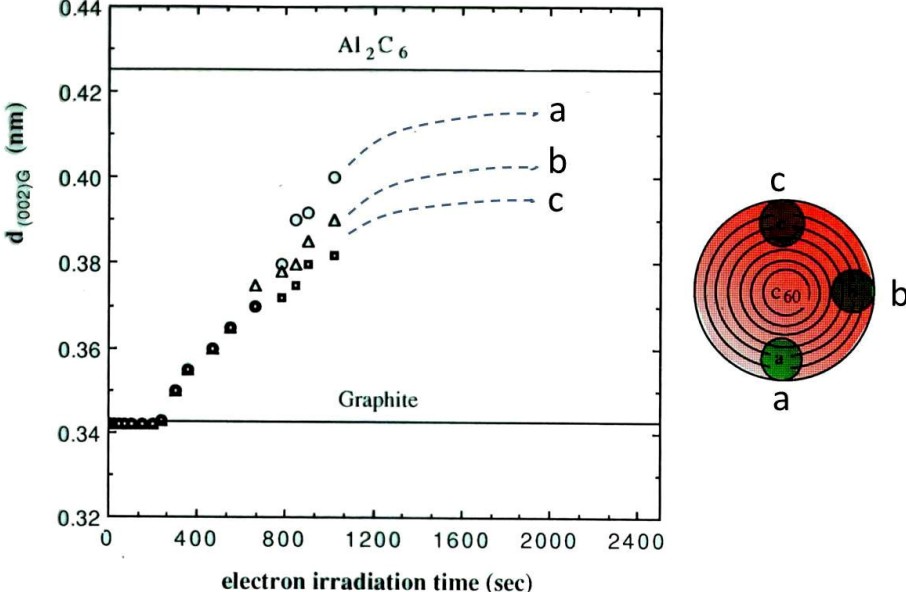

**Figure 14.** Expansion of the onion-like graphite lattice spacing $d_{(002)G}$ due to Al intercalation, under electron beam irradiation of $1.0 \times 10^{20}$ e/cm$^2$s. The spacing increased with electron irradiation time and seemed to saturate at 0.425 nm, which coincides with the spacing of the compound $Al_2C_6$. The limit of lattice expansion stems either from the blocking effect due to the coexistence of Cu atoms or is constrained by multiple carbon layers [11].

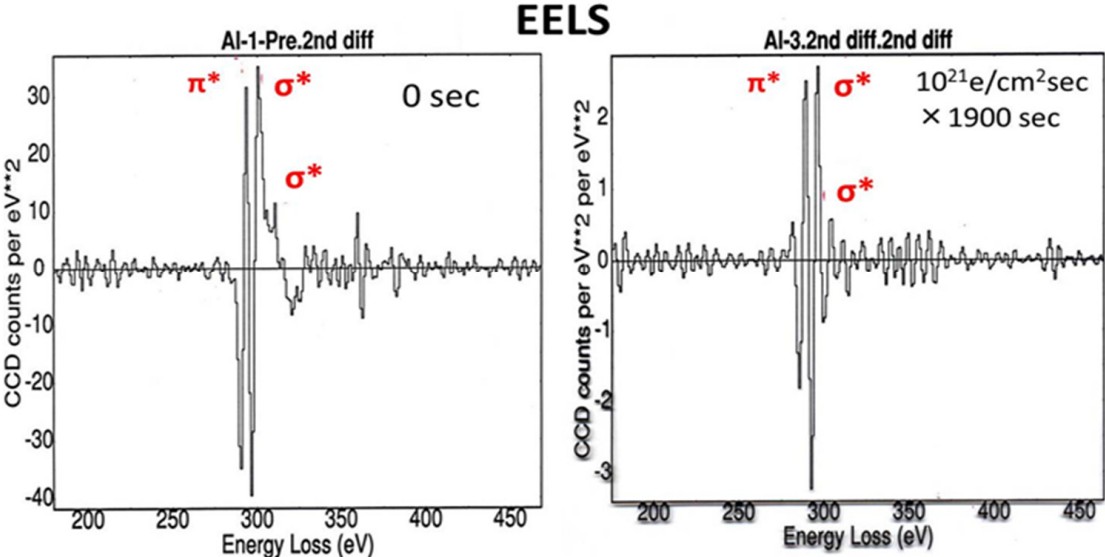

**Figure 15.** Second derivative of the EELS of the intercalated structure formed under electron irradiation of $1.0 \times 10^{21}$ e/cm$^2$s for 1900 s. No change is observed in the π-bond between layers after irradiation, whereas the σ-bond, sp$^2$ in-plane, decreased, which suggests the Al and Cu intercalated atoms partially replaced the carbon atoms [12].

## 4. Al$_2$O$_3$ Derivatives

Continuous electron irradiation of metastable θ-Al$_2$O$_3$ at $2.1 \times 10^{20}$ e/cm$^2$s produced Al nanoparticles and stable α-Al$_2$O$_3$ particles, as shown in Figure 2. When the intensity and area of electron irradiation changed abruptly (i.e., flashing of the beam), other-shaped nanosized aluminium oxides were obtained. In this section, I present the oxide derivatives of Al$_2$O$_3$ oxides with different shapes and phases.

### 4.1. α-Al$_2$O$_3$ Nanorods

Although α-Al$_2$O$_3$ with a polygonal shape was deposited beside θ-Al$_2$O$_3$ by the recombination of Al and O atoms, the α-Al$_2$O$_3$ nanorods shown in Figure 16 grew from the surface of the parent α-Al$_2$O$_3$. The rods had a faceted structure surrounded by {100} surfaces and grew in the {110} direction. Figure 16 was obtained in situ, in which the nanorods grew without an amorphous oxide film.

### 4.2. δ-, θ-Al$_2$O$_3$ Nanoballs/Nanowires

When metastable Al$_2$O$_3$ was irradiated by electrons with a higher density over a short time, the structure was retained, but a different nanostructure was also obtained. Kameyama and Tanaka abruptly switched between intensities of $5.5 \times 10^{22}$ and $5 \times 10^{19}$ e/cm$^2$s in a so-called "flashing mode," maintaining each state for 0.1 s, which facilitated the formation of θ-, δ-Al$_2$O$_3$ nanorods/nanoballs, as shown in Figure 17, respectively [13].

Both θ-Al$_2$O$_3$ nanowires and nanoballs grown using flashing-mode electron beam irradiation are shown in Figure 17. They connected and grew toward the irradiation centre from the original θ-Al$_2$O$_3$ particle surface. The nanowire and nanoball had a (111)//(101) epitaxial relationship, as shown in Figure 18. The θ-Al$_2$O$_3$ nanoball was a sphere of 30 nm in diameter covered with an amorphous Al-O layer, suggesting that the impact of a higher irradiance electron beam resulted in a temperature rise of approximately 400 °C, reaction, and rapid cooling without phase transformation. The δ-Al$_2$O$_3$ nanowires and nanoballs shown in Figure 19 were obtained by applying the same flashing electron beam to δ-Al$_2$O$_3$ particles. The diameter of the δ-Al$_2$O$_3$ particles was 20 nm, which was slightly smaller than that of the θ-Al$_2$O$_3$ particles, and no explicit epitaxy relation was observed.

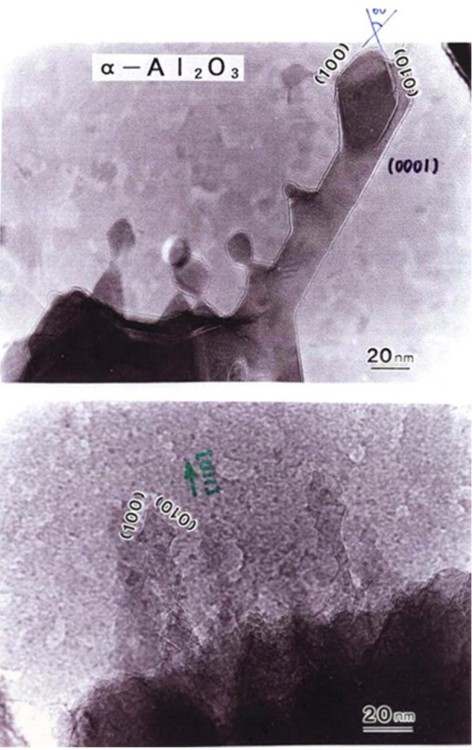

**Figure 16.** $\alpha$-$Al_2O_3$ rods epitaxially grown from the $\theta$-$Al_2O_3$ surface by electron beam irradiation at $2.1 \times 10^{20}$ e/cm$^2$s, as shown in Figure 2. The rods grew in the {110} direction and were faceted by {100} surfaces [3].

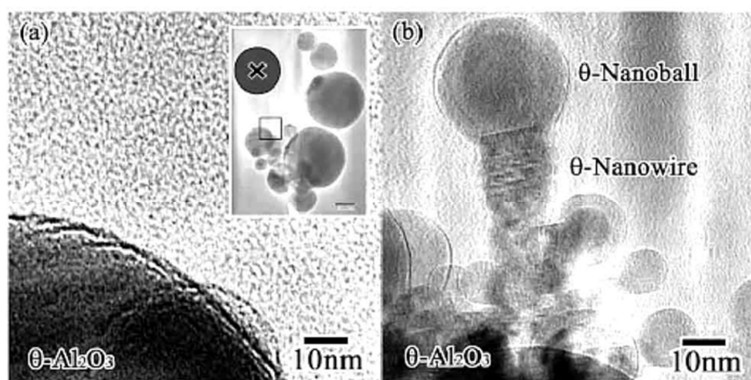

**Figure 17.** $\theta$-$Al_2O_3$ nanowire and nanoball grown under flashing mode electron beam irradiation (i.e., rapid switching between intensities of $5.5 \times 10^{22}$ and $5 \times 10^{19}$ e/cm$^2$s). They connected and grew toward the irradiation centre from the original $\theta$-$Al_2O_3$ particle surface. The nanowire grew epitaxially maintaining the same plane as the parent $\theta$-$Al_2O_3$ particle. (**a**) $\theta$-$Al_2O_3$ particles before irradiation, with X indicating the irradiation centre. (**b**) $\theta$-$Al_2O_3$ nanowire and nanoball grown after electron beam irradiation [13].

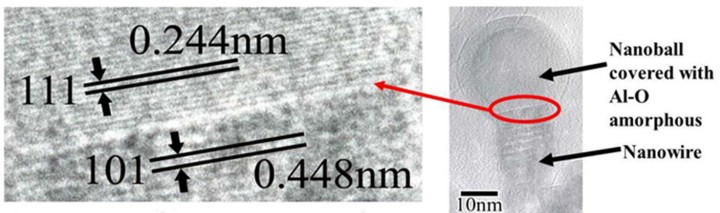

**Figure 18.** The interface of the $\theta$-$Al_2O_3$ nanowire and nanoball, shown in Figure 17, exhibits a (111)//(101) epitaxial relationship. The nanoball is covered with an amorphous Al-O layer [14].

### 4.3. Al$_2$O$_3$ Nanoparticle Encapsulation

The surface of the Al$_2$O$_3$ particles was easily covered by an amorphous hydrocarbon layer through a reaction with residual gases in the TEM, where CO, H$_2$O, and H$_2$ remained even in a highly evacuated atmosphere. When the particles were irradiated by electrons through the outer layer, their inner volume was pulverised into smaller nanoparticles. An example is shown in Figure 20, for the case in which δ-Al$_2$O$_3$ particles with 200 nm in diameter were transformed into δ- and α-Al$_2$O$_3$ nanoballs with 2–20 nm in diameter, encapsulated by a hydrocarbon skin. This structure was generated by irradiation with an intensity of $1.6 \times 10^{20}$ e/cm$^2$s, which was as high as that used for nanoparticle preparation and manipulation. Nanoparticle encapsulation technology may be applicable to drug delivery systems in medicine [15].

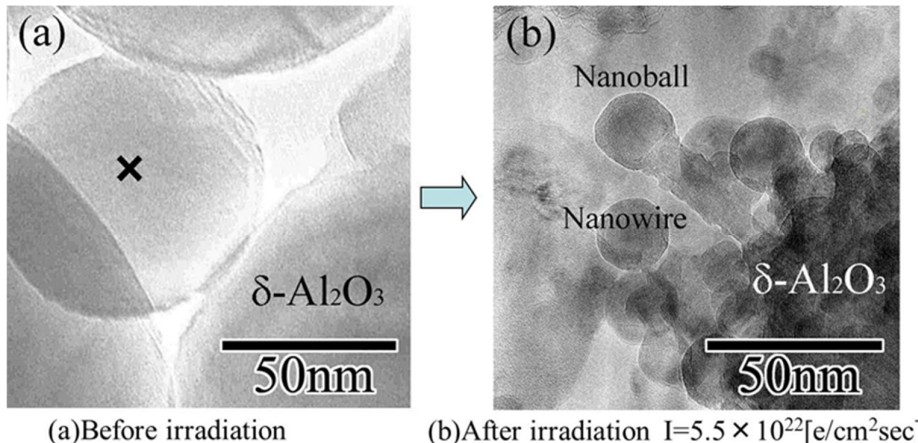

**Figure 19.** δ-Al$_2$O$_3$ nanowire and nanoball grown under flashing mode electron beam irradiation with intensities switching between $5.5 \times 10^{22}$ and $5 \times 10^{19}$ e/cm$^2$s. They connected and grew toward the irradiation centre from the original δ-Al$_2$O$_3$ particle surface. (**a**) δ-Al$_2$O$_3$ particles before irradiation, with X indicating the irradiation centre. (**b**) δ-Al$_2$O$_3$ nanowire and nanoball grown after electron beam irradiation [13].

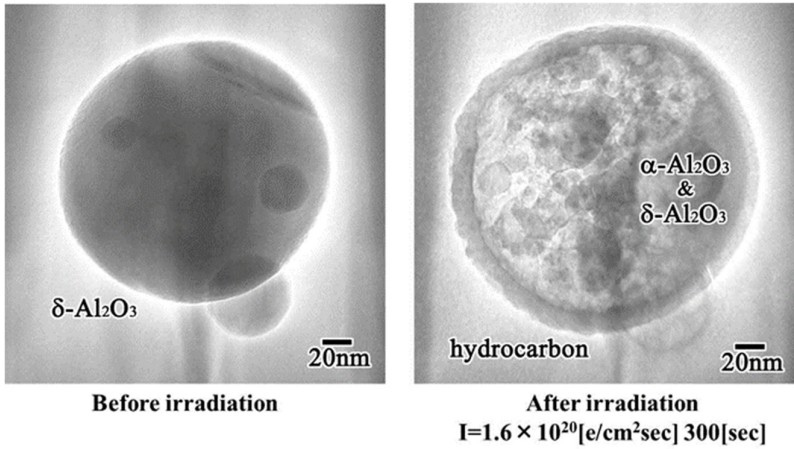

**Figure 20.** δ- and α-Al$_2$O$_3$ nanoball-encapsulated structures obtained from δ-Al$_2$O$_3$ particles by electron beam irradiation at $1.6 \times 10^{20}$ e/cm$^2$s for 300 s. An outer amorphous hydrocarbon layer is formed from residual gas contaminants in TEM [15].

## 5. Other Nanoparticles

### 5.1. W Nanoparticles and Manipulation

In Section 2, a novel method for preparing oxide-free Al nanoparticles from metastable oxides using electron beam irradiation was discussed. This method can be extended to other nanoparticles of easy oxide-forming elements, such as W. W has a heavier specific

weight of 19.3 g/cm$^3$, and W-O has a larger bonding enthalpy such that a higher electron irradiation intensity is required to obtain W nanoparticles from WO$_3$. Although the fundamental electron optics in the TEM were the same as those used for Al nanoparticles, electrons from the field emission source provided a higher intensity of 10$^{23}$ e/cm$^2$s than the 10$^{20}$e/cm$^2$s obtained from the LaB$_6$ filament. Using a Hitachi HF-2000 TEM equipped with a field emission gun with an intensity of 4 × 10$^{23}$ e/cm$^2$s, Tamou and Tanaka reported the formation of W nanoparticles with an average diameter of 4.3 nm, as shown in Figure 21. The EELS spectra showed no oxygen atoms on the W surface, as shown in Figure 22 [16].

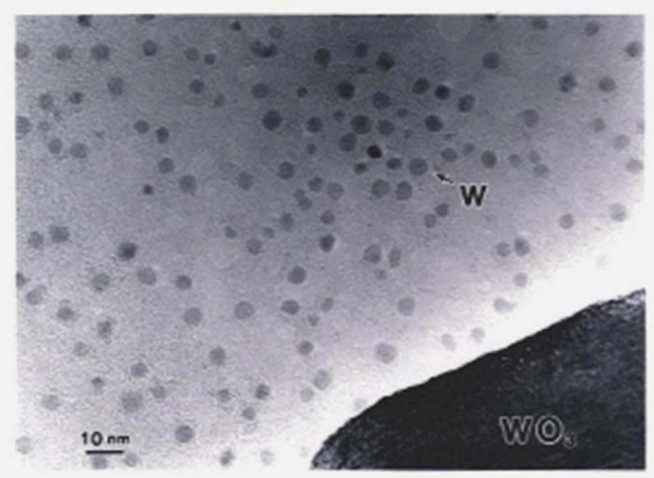

**Figure 21.** W nanoparticles deposited by electron beam irradiation of a WO$_3$ particle at 4 × 10$^{23}$ e/cm$^2$s (6 × 10$^8$ A/m$^2$) for a few seconds. The diameter of the W particles ranged between 2 and 6 nm with an average of 4.3 nm [16].

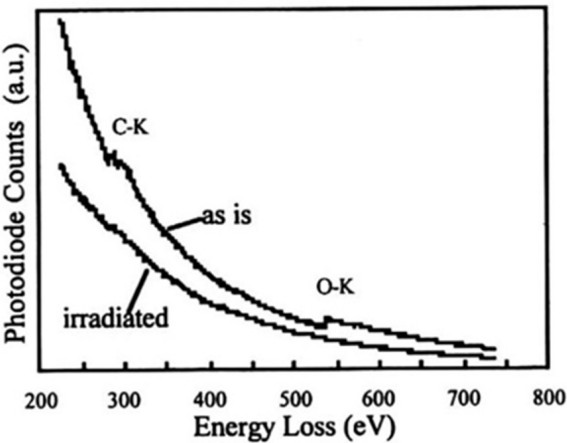

**Figure 22.** EELS spectra before and after electron irradiation of WO$_3$. Electron irradiation induced the disappearance of the O-K$\alpha$ peak to form pure W nanoparticles [16].

### 5.2. W Migration to Bond and Fullerene Formation

Further electron irradiation of two W nanoparticles, obtained as in Figure 21 at 1.9 × 10$^{21}$ e/cm$^2$s, which is an irradiation 10 times higher than that used to form Al as shown in Figure 6, induced migration, bonding, and coalescence, as shown in Figure 23 [16]. Graphitic shells also nucleated beneath the W nanoparticles from the amorphous carbon film and grew to onion-like fullerene, which is the same phenomenon observed with Al nanoparticles shown in Figures 10–12 [16].

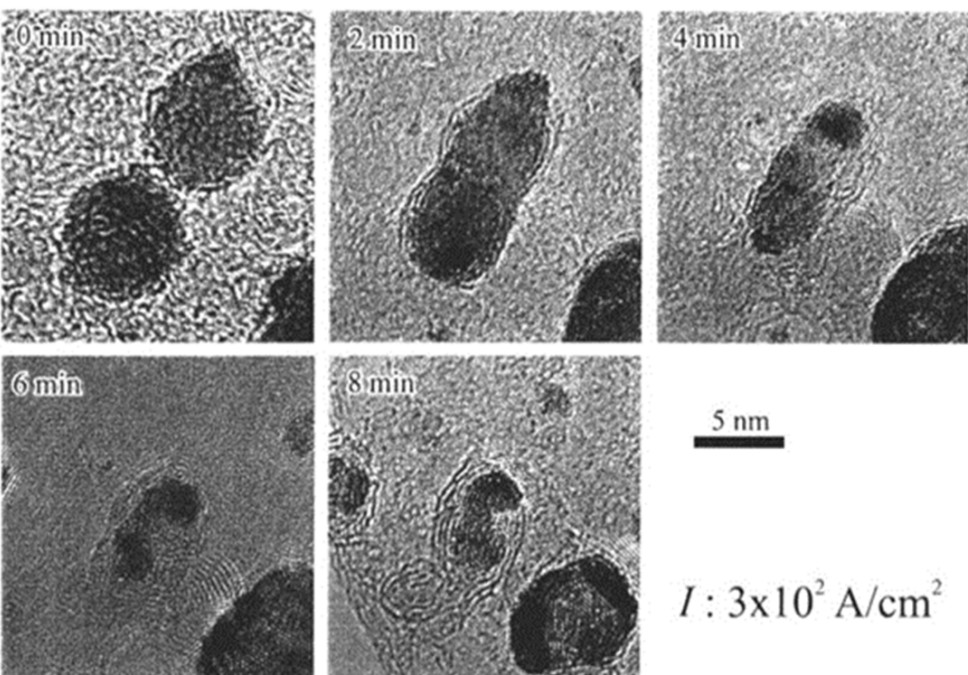

**Figure 23.** Effects of electron beam irradiation of W nanoparticles on an amorphous carbon film at $1.9 \times 10^{21}$ e/cm²s (300 A/cm²). W nanoparticles migrated together and coalesced, followed by fullerene formation between the W particles and the carbon film. The general features of nanostructure evolution were the same as those observed with irradiation of Al nanoparticles shown in Figure 10 [16].

### 5.3. Bonding of Pt and Cu Nanoparticles

Electron irradiation of a group of Pt and Cu nanoparticles induced bonding to form nanofilms, as shown in Figures 24–26. In these cases, nanoparticles were prepared by Ar ion sputtering with a diameter of 10 nm for Pt and 50 nm for Cu. The irradiation intensity for Pt was the same as that for Al, whereas it was 100 times higher for Cu. Bonded Pt/Pt mainly showed three stable $\Sigma 3$ twin boundaries. The Cu particles migrated to the irradiation centre and bonded, as shown in Figures 26 and 27. The driving force was also the momentum transfer from electrons in the pole piece of the TEM, as shown in Figures 4 and 5 [17–20].

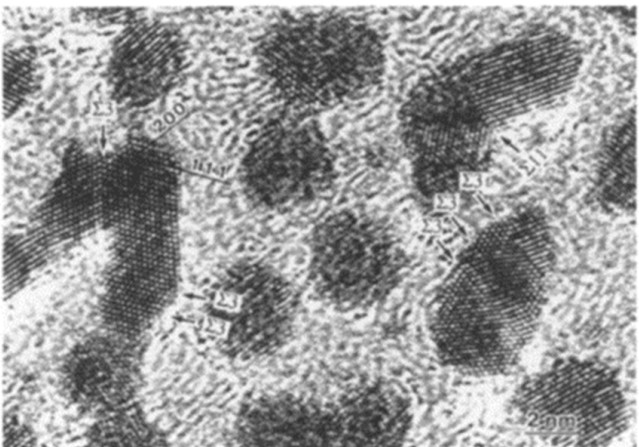

**Figure 24.** Pt nanoparticles bonded by electron irradiation with an intensity of $2.1 \times 10^{20}$ e/cm²s for 700 s. The bonded Pt/Pt nanoparticles had tilt boundaries of $\Sigma 3$ and $\Sigma 11$ [19].

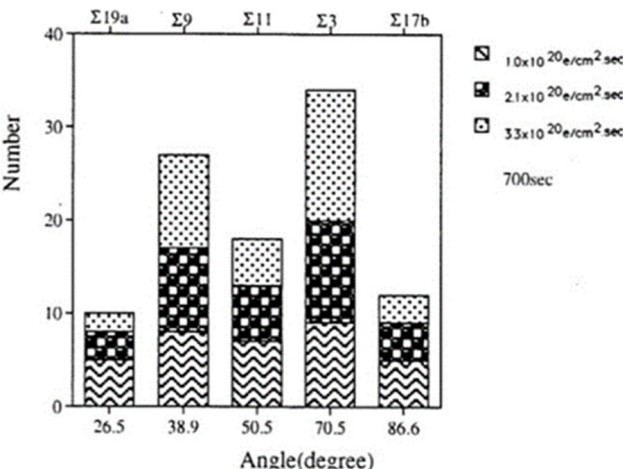

**Figure 25.** Histogram of tilt boundaries under electron irradiation at three intensities from 1.0 to $3.3 \times 10^{20}$ e/cm²s for 700 s. Σ3 CSL boundaries were predominant as a low energy structure [19].

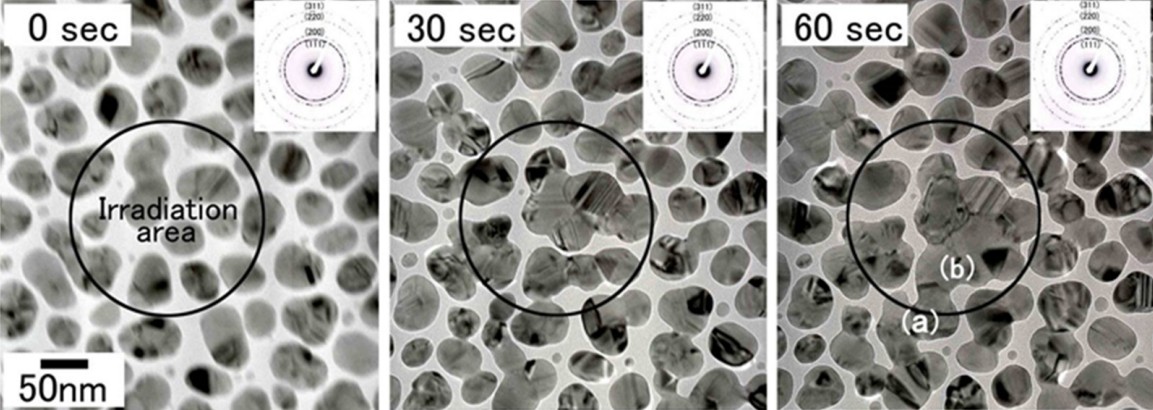

**Figure 26.** Change of the bright-field images and electron diffraction patterns of Cu nanoparticles irradiated with electrons at an intensity of $5.5 \times 10^{22}$ e/cm²s for 60 s. Cu nanoparticles migrated to the irradiation centre and bonded with each other in the marked irradiation area. The electron diffraction patterns, typical Cu Debye rings, did not change during irradiation. The nanostructures of the bonded interface (i.e., CSL) boundary, were obtained in regions (**a**) and (**b**) after 60 s of irradiation, as shown in Figure 27 [20].

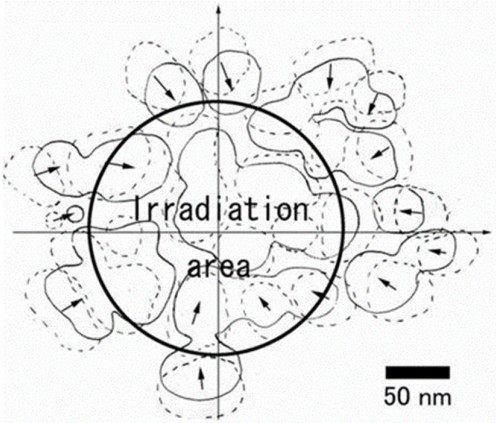

**Figure 27.** Superposed view of Cu nanoparticle migration and bonding during the irradiation times of 0 s (dotted line) and 30 s (solid line). The circle indicates the electron beam irradiation area of 200 nm diameter. Nanoparticles migrated toward the irradiation centre and finally bonded together. Unbonded small nanoparticles seemed to revolve clockwise around the irradiation centre [20].

## 6. Nature of Nanoparticle Manipulation and Nanostructure Modification by Electron Beam Irradiation

*6.1. Temperature Rise in Al Nanoparticle Manipulation*

In Section 3.2, various types of manipulation of Al nanoparticles on the TEM specimen stage were explained. These were migration, bonding, rotation, revolution, and embedding of the nanoparticles, and the driving forces were explained as tangential and centripetal forces, as shown in Figure 4. Another possibility of manipulation is the temperature rise caused by electron irradiation to induce their movement. Xu and Tanaka [10] estimated the temperature rise at the stage as 10° C at most, based on Equation (1) using Fisher's theory [21]:

$$Tm - Tg = r^2 I_0 \Delta E [a_0 + \ln(R/r)^2]/4kz, \tag{1}$$

where Tm is the maximum temperature of the carbon film, Tg is the temperature of the Cu support grid, namely Tm − Tg is the temperature rise by electron beam irradiation, r is the radius of the irradiation beam, $I_0$ is the intensity of the irradiation beam ($10^{20}$ e/cm$^2$s), $\Delta E$ is the energy loss of the incident electron in the carbon film, when it is <1000 nm thick, $a_0$ is Euler's constant (0.5772), R is the distance between the irradiation beam centre and the Cu grid bar, k is the thermal conductivity of carbon, and z is the thickness of the carbon film (20 nm).

The heating effect in the localised area under the irradiation condition of $10^{20}$ e/cm$^2$s can be a minor effect, and the Lorenz force or the momentum transfer from electrons and ionised atoms is the major effect of the manipulation. This effect is clearly supported by the counterclockwise revolution caused by the magnetic field change, as shown in Figure 8 [8].

*6.2. Temperature Rise in Al$_2$O$_3$ Nanocomplex and W Nanoparticles*

When θ-Al$_2$O$_3$ was irradiated at a density of $10^{19}$–$10^{20}$ e/cm$^2$s, Al and α-Al$_2$O$_3$ were formed, as shown in Figures 2 and 16, where the reaction proceeded with a small temperature increase of the order of 10 °C, as shown in Section 6.1. On the contrary, the flashing mode of electron irradiation by rapid switching between intensities of $5.5 \times 10^{22}$ and $5 \times 10^{19}$ e/cm$^2$s was applied to θ- and δ-Al$_2$O$_3$ to induce Al$_2$O$_3$ nanoball/nanowire complexes, as shown in Figures 17 and 19 [13]. The higher electron beam intensity increased the temperature by more than 300 °C, as calculated through $I_0$ in Equation (1) by maintaining $5.5 \times 10^{22}$ e/cm$^2$s even in a short time of less than 0.1 s. This temperature increase was also predicted by Yokota et al. [22]. Rapid and concentrated heat input at the localised resulted in an Al-O recombined nanoball/nanowire complex with epitaxy at the interface, as shown in Figure 18 [14].

Heavy atoms such as W required a higher irradiation intensity of $4 \times 10^{23}$ e/cm$^2$s to obtain W nanoparticles, as shown in Figure 21 [16]. Although the binding energy of W–O in the starting material WO$_3$ was smaller than that of Al–O, the W atom is ten times heavier than the Al atom, and required a higher energy for sputtering. A temperature rise was also expected in this irradiation condition, but no melting was observed because of its higher melting point, 3680 K.

*6.3. Lorentz Force in Nanoparticle Manipulation*

To discuss the mechanism of nanoparticle manipulation in a TEM, the interaction between electrons and nanoparticles on the specimen stage in the magnetic field was analysed. The TEM used in this study was 200 keV JEM-2010, which has a pole piece with a magnetic field of $10^4$ Gauss from top to bottom, where the specimen stage is located slightly above the centre plane. In Figure 4, the electron trajectory is illustrated schematically by Horiuchi et al. [4], and the Lorentz forces F$_1$ and F$_2$ arise from the magnetic components Br and Bz, respectively, with spirally running electrons. The specimen stage is located between planes 1 and 2, and Al, W, Pt, and Cu nanoparticles experience both a tangential force to rotate and revolve and a centripetal force to migrate, bond,

and embed. The momentum transfer from electrons to nanoparticles is the source of this movement.

The Lorentz force exerted on one electron, $F_e$, was roughly estimated by Equation (2)

$$F_e = m_e v_e^2 / r, \tag{2}$$

where $m_e$ is the mass of one static electron as $9.1094 \times 10^{-31}$ kg, $v_e$ is the velocity of electrons, considering the relativistic effects at 200 kV as $v_e = v_{200}$ 1.3914 = $2.900 \times 10^8$ m/s, and r is the distance between the nanoparticle and the irradiation centre. When assuming the experimental case of Al nanoparticles shown in Figure 5, r = 60 nm, the Lorentz force from one electron $F_e$ was $1.277 \times 10^{-6}$ N. The total Lorentz force, F, to the Al nanoparticle of 20 nm in diameter with an irradiation time of 1200 s at $10^{20}$ e/cm$^2$s was estimated to be $9.63 \times 10^5$ N. Although this is the maximum value, which occurs when electrons travel from plane 1 to 2 and the driving force of nanoparticle manipulation changes the direction from tangential to centripetal, it is too high for nanoparticle movement. The author proposes the following reasons: although the electron density was measured on the fluorescent plate beneath the stage, accelerated electron velocity decreased while travelling inside a TEM, and electrons lost their kinetic energy through ionisation of the wall by their impact. The Lorentz force decreased by at least 1/100. The existence of a friction force between nanoparticles and a substrate carbon film could also be one of the causes. The effective cross-section of the nanoparticle might be considered, which decreases the impact of the electrons.

The Lorentz force for nanoparticle manipulation is also valid for W, Pt, and Cu, as shown in Figures 23, 24, and 26. Although the time to bond is different depending on the density of the weight, electron irradiation focusing to the localised region will be a candidate technology for fabricating circuits or functional dots by nanoparticle arrays.

## 7. Summary of the Nanostructure Evolution and Manipulations in the Electron Excited Field

Research conducted by my group on nanostructure evolution by electron beam irradiation from 1995 to 2005 was reviewed. I have utilised electron beams in TEM to synthesise nanomaterials and manipulate their nanostructures, in addition to observing and analysing nanostructures. An overview of the effects of electron irradiation is presented in Figures 28 and 29, where the abscissa is expressed as the electron irradiation intensity on a logarithmic scale. The electron beam was focused for synthesis and manipulation up to $10^{19}$–$10^{24}$ e/cm$^2$s, which is higher than $10^{16}$ e/cm$^2$s generally used for electron diffraction and $10^{18}$ e/cm$^2$s used for bright-field imaging.

Figure 28 shows that electron irradiation of metastable θ-$Al_2O_3$ provides oxide-free Al nanoparticles, rod-like α-$Al_2O_3$, and encapsulated nanoparticles, whereas flashing mode provides θ-, δ-$Al_2O_3$ nanoball/nanowire complexes. The formation of W nanoparticles from $WO_3$ requires a higher intensity of more than $10^{23}$ e/cm$^2$s. Electrons traveling in a spiral trajectory in the magnetic field of the pole piece transfer momentum to the Al nanoparticles enable various types of manipulation, such as migration, bonding, rotation, revolution, embedding, fullerene formation, and intercalation. The intensity is also more than 100 times higher than that of normal observation conditions, as shown in Figure 29. The combination of such syntheses and manipulation will provide more complicated nanostructures for future applications.

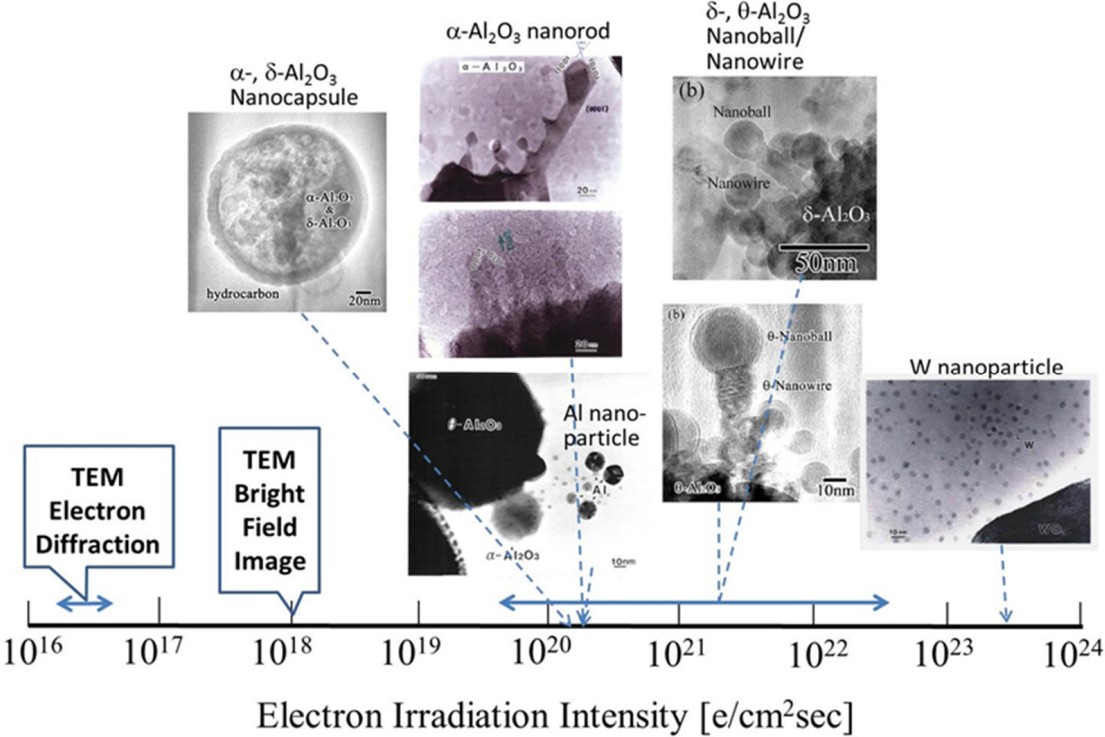

**Figure 28.** Nanostructured materials obtained by electron irradiation in TEM. Nanoparticles and nanosized oxides can be induced in an electron excited reaction field. The electron irradiation intensity ranged from $10^{20}$–$10^{23}$ e/cm$^2$s depending on the specific gravity of the materials and the metal-oxygen binding enthalpy of the starting oxide.

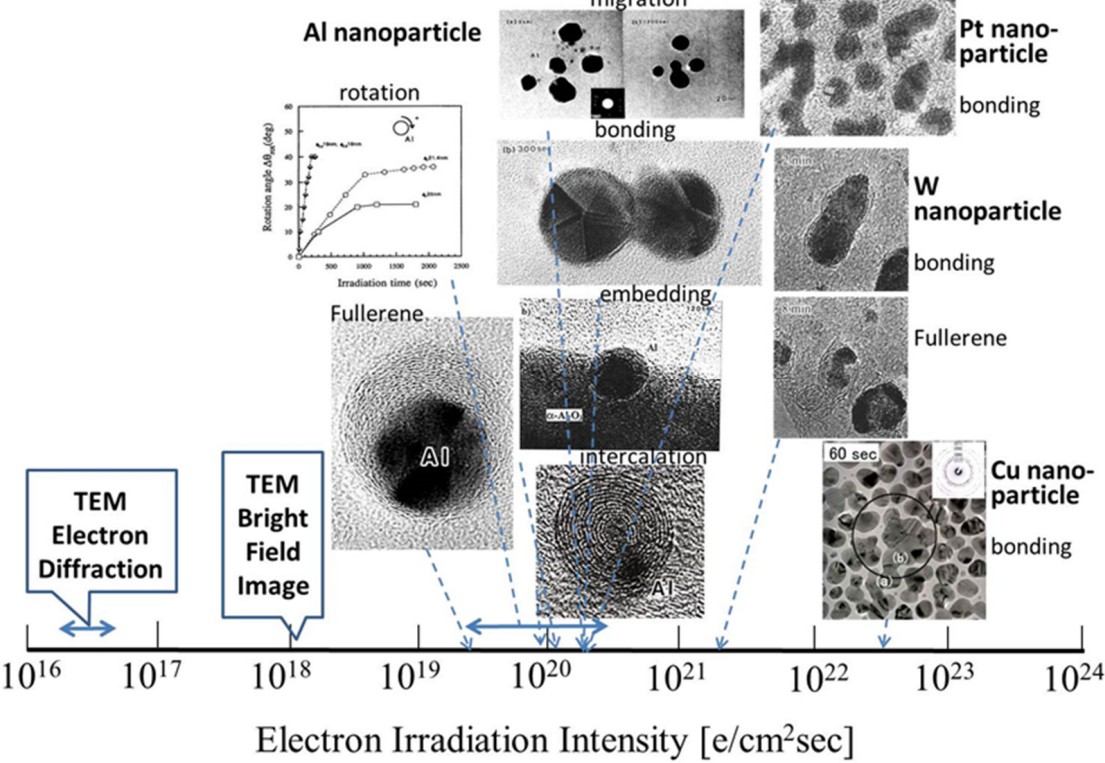

**Figure 29.** Manipulation of nanomaterials by electron irradiation in a TEM. Nanostructures can be controlled in an electron excited reaction field through migration, bonding, rotation, revolution, embedding, fullerene formation and intercalation. The electron irradiation intensity ranged between $10^{19}$–$10^{23}$ e/cm$^2$s depending on the size and weight of the material.

## 8. Recent Development in Control and Manipulation of Nanostructured Materials

In this review, pioneering works by the author's group published in 1995–2005 are summarised as a tool for nanomaterial control and manipulation at the TEM room temperature stage. In these works, mediate-accelerating keV was initially used, followed by accompanying magnetic field, and focusing electrons to the localised area. Although there are several works on the effects of electron irradiation, reviews on this topic are scarce. For example, lattice defects such as point defects or stacking faults are introduced as radiation damage in the region of MeV electron irradiation, for which an ultra-high voltage TEM has been used as an experimental simulation. Krasheninnikov et al. published an excellent review paper on the effects of ion and electron irradiation, collecting more than 680 papers [23], which contained derivation and simulation for nanostructured materials. Accompanying magnetic field and focusing electrons to the localised area in TEM are unique technologies for manipulation, which were partly covered in the papers by Zheng et al. [24] and Andres et al. [25]. Zheng et al. reported that the trapping force for one nanoparticle was on the order of $10^{-9}$ N in the electron density gradient of $10^{18-19}$ e/cm$^2$s [24] which is the same order of magnitude as discussed in Section 6.3. Simulation by first-principle theory using the density of state is important for predicting the formation, growth, and coalescence of nanoparticles [25].

## 9. Conclusions

The author's group succeeded in inducing the formation of Al nanoparticles by electron irradiation of metastable θ-Al$_2$O$_3$, followed by manipulation of the nanoparticles. A series of phenomena was observed without heating using high-resolution TEM (HRTEM), with an electron beam intensity as low as $10^{19-20}$ e/cm$^2$s. The typical morphology of the nanoparticles was that of a nanodecahedron surrounded by (111) surfaces with twins. Electron beam irradiation of a group of Al nanoparticles promoted their rotation, revolution, and migration to the irradiation centre, resulting in bonding and embedding into an α-Al$_2$O$_3$ matrix. The driving force is considered to be the momentum transfer from electrons spiralling across the pole piece of the HRTEM in a strong magnetic field to the Al nanoparticles. When nanoparticles were placed on an amorphous carbon film, onion-like fullerene nucleated and grew beneath them, and finally, a metallofullerene or Al-atom-intercalated structure was formed by electron irradiation.

To develop the manipulation technology for other types of nanoparticles, an electron beam was used to irradiate Cu nanoparticles of 10–50 nm in diameter at an irradiation intensity of $5.5 \times 10^{22}$ e/cm$^2$s, Pt nanoparticles at $1.0–3.3 \times 10^{20}$ e/cm$^2$s, and W nanoparticles derived from WO$_3$ at $9 \times 10^{20}$ to $4 \times 10^{23}$ e/cm$^2$s. The behaviour of Cu, Pt, and W nanoparticles under electron irradiation was similar to that of Al, and a nanofilm was finally formed. The CSL boundary structures at the bonded interface of Cu nanoparticles were found to be unstable Σ7 and Σ13b, which are different from the stable Σ3 obtained in Al and Pt with weaker electron beam irradiation.

The possible scientific contribution of electron irradiation is the synthesis of materials in a metastable state through a nonequilibrium reaction in vacuum, as well as the induction of hybridised nano-/mesostructures. It also enables the study of the nature of materials in a pristine and controlled environment, for example, without the formation of an oxide. From the viewpoint of application to devices, nanosized balls, dots, wires, and tube-forming three-dimensional structured circuits may be used as elements of nanodevices, and chemically active points embedded in the substrate for use as a catalyst can be achieved by the manipulation of electron irradiation. With respect to industrial applications, our technologies will contribute to the development of micro- and nanoelectromechanical systems, memories, photonics, battery electrodes, H$_2$ storage, and more.

**Funding:** This research partly received funding from ERATO, JST.

**Acknowledgments:** The author gratefully acknowledges the research scientists at Tanaka Solid Junction Project, JST, BingShe Xu (now: Shaanxi Univ. Sci. Tech.) and Yoshitaka Tamou (now Mitsubishi Materials), and graduate students at Nagoya Institute of Technology, Toru Kameyama (now: Sony GMO Corp.) and Yoshiki Miki (now Nippon Giant Tire Co., Ltd.).

**Conflicts of Interest:** The author declares no conflict of interest.

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
