# Peer review of "Control and Modification of Nanostructured Materials by Electron Beam Irradiation"

_qubs, doi:10.3390/qubs5030023_

Round 1
Reviewer 1 Report
The article provides an overview of experimental results on the production of nanoscale (3 nm) aluminum particles from aluminum oxides under the influence of electron beams. These results have been obtained for more than 20 years, but have not lost their importance at the present time, due to the possibility of obtaining nanoscale elements for MEMS technologies. Therefore, to attract the attention of modern young scientists, this article is extremely useful.
As a wish. It would be good to give estimates of the energies: the Al-O bond, the electrons in the beam. It is desirable to describe the mechanism of the reaction field.
Author Response
Dear Prof. Reviewer 1,
Based on your comments, I have brushed up English and added intensive consideration to the first manuscript. Discussion appearing in new Sections is listed as follows:
- In Section 3.1, I have added discussion on metastable phases of aluminum oxide from viewpoints of the formation energy.
- In new Section 6, intensive consideration on the nature of nanoparticle manipulation and nanostructure modification by electron beam irradiation. It consists of temperature rise in 6.1 and 6.2 and the Lorenz force for manipulation in 6.3.
- In new section 8, papers recently published are reviewed which closely related with my concepts.

Reviewer 2 Report
This paper summarizes work conducted and published 15-20 years ago on materials modification by high intensity electron beam from a TEM. The results are interesting and many examples are shown of nanostructures formed and modified during beam exposure, but there is no sincere discussion on the mechanism or physics/chemistry behind the transformations. The only mechanism that is mentioned is “momentum transfer from electrons”, but there is no further discussion on, for instance, flux versus fluence, themal effects, charge state, or influence of the electron energy. Such an attempt to understand the processes would have added substantial value to the present review.
There is some interest in documenting and summarizing all the different results discovered two decades ago and maybe the manuscript can be published. There are, however, issues with the English and several typos that need to be sorted out.
Furthermore, I think the energy of the electrons should also be stated for the different examples shown, if someone should be more interested in understanding the mechanism in greater detail.
The reference list contains 19 references of which the author involved in nearly all. This does not strengthen the impression of the article. I can understand that the field is rather limited, but a quick Google search gave at least 10 other authors with fair citation numbers that have published more recently in this field. Please, try to include more references by other authors.
Author Response
Dear Prof. Reviewer 2,
Based on your comments, I have brushed up English and style, added intensive consideration to the first manuscript to reply to your Judge "Major Revision"
Discussion appearing as new Sections or sentences is listed as follows:
- In Section 3.1, I have added discussion on metastable phases of aluminium oxide from viewpoints of the formation energy.
- In new Section 6, intensive consideration on the nature of nanoparticle manipulation and nanostructure modification by electron beam irradiation. It consists of temperature rise in 6.1 and 6.2 and the Lorentz force for manipulation in 6.3.
- In new Section 8, papers recently published are reviewed which closely related with my concepts.

Round 2
Reviewer 2 Report
The paper is now ready for publication.